# Understanding the Contribution of Lactate Metabolism in Cancer Progress: A Perspective from Isomers

**DOI:** 10.3390/cancers15010087

**Published:** 2022-12-23

**Authors:** Ming Cai, Jian Wan, Keren Cai, Haihan Song, Yujiao Wang, Wanju Sun, Jingyun Hu

**Affiliations:** 1College of Rehabilitation Sciences, Shanghai University of Medicine and Health Sciences, Shanghai 201318, China; 2Department of Emergency and Critical Care Medicine, Shanghai Pudong New Area People’s Hospital, Shanghai 201299, China; 3Central Lab, Shanghai Key Laboratory of Pathogenic Fungi Medical Testing, Shanghai Pudong New Area People’s Hospital, Shanghai 201299, China; 4Department of Rehabilitation Medicine, Shanghai Pudong New Area People’s Hospital, Shanghai 201299, China

**Keywords:** lactate metabolism, L-lactate, D-lactate, cancer progression, anticancer therapy

## Abstract

**Simple Summary:**

Lactate (L-lactate and D-lactate) is the main production of the Warburg effect, which is vital for carcinoma cell metabolism. This review retrospects the lactate isomer metabolism in the cancer progress. The related enzyme and proteins have been listed as prognostic biomarkers for cancers, and the lactate down-streamed molecular cancerogenic signaling is also introduced. This review will provide a new strategy for anticancer therapy that targets lactate metabolism.

**Abstract:**

Lactate mediates multiple cell-intrinsic effects in cancer metabolism in terms of development, maintenance, and metastasis and is often correlated with poor prognosis. Its functions are undertaken as an energy source for neighboring carcinoma cells and serve as a lactormone for oncogenic signaling pathways. Indeed, two isomers of lactate are produced in the Warburg effect: L-lactate and D-lactate. L-lactate is the main end-production of glycolytic fermentation which catalyzes glucose, and tiny D-lactate is fabricated through the glyoxalase system. Their production inevitably affects cancer development and therapy. Here, we systematically review the mechanisms of lactate isomers production, and highlight emerging evidence of the carcinogenic biological effects of lactate and its isomers in cancer. Accordingly, therapy that targets lactate and its metabolism is a promising approach for anticancer treatment.

## 1. Introduction

The Warburg effect describes a unique phenomenon that cancers incline to shift the mode of oxidative phosphorylation (OXPHOS) to glycolysis in spite of abundant oxygen [1,2]. Lactate is the main production of glycolysis [3], which contains two isomers, L-lactate and D-lactate. The accumulation of high lactate in solid tumors and its extracellular environment is considered as the key and early evidence of malignant development, which is associated with a poor prognosis [4,5]. Lactate reprograms the tumor microenvironment (TME) to have profound effects on cancer cell phenotype [6,7] and is conducive to the progress of cancer that involves the eight biological capabilities acquired of cancer: sustaining cell proliferation, promoting growth, resisting cell death, enabling replicative immortality, inducing angiogenesis, activating invasion and metastasis, reprogramming energy metabolism, and evading immune destruction [8]. Lactate’s contribution to cancer is not only the respiratory fuel [3] but also the regulator of intracellular and extracellular molecular signaling in the TME.

In the current review, we describe the link of L- and D-lactate production with aerobic glycolysis in detail. We also discuss the current advances of lactate in cancer, including breast, cervical, lung, pancreatic, prostate, and liver cancer, and focus on the role of two lactate isomers in the cancer progress. As discussed in the review, lactate-related prognostic markers in cancer and downstream molecular signaling are concerned with a better understanding of lactate metabolism. The lactate effect will increasingly influence the development of new cancer treatments and strategies to overcome resistance to existing treatments.

## 2. Lactate Metabolism in Carcinoma Cells

### 2.1. Warburg Effect

The Warburg effect is an extremely common event in many carcinoma cells [9]. This amazing theory was firstly proposed by Otto Warburg and colleagues in the 1920s [10], which has been documented for over 100 years [11]. It describes the unusual metabolic transforming phenomenon in carcinoma cells that, unlike most normal tissues, carcinoma cells tend to metabolize most glucose into lactate for adenosine triphosphate (ATP) production even in the presence of sufficient oxygen, which is termed “aerobic glycolysis” [1,2]. It is not the defective ability of mitochondrial oxidative phosphorylation (OXPHOS) in carcinoma cells leading to no alternative choice. On the contrary, the mitochondrial function is intact [12,13,14], even perhaps with higher-efficiency of OXPHOS in carcinoma cell types [15,16]. In fact, there are several potential advantages of glycolysis in carcinoma cells. For example, glycolysis can provide energy supply more rapidly than the aerobic oxidation for the proliferation of carcinoma cells in spite of less efficient ATP production in this way [11]. Glycolysis reduces the reliance on oxygen for ATP production and thereby, the potentially destructive reactive oxygen species (ROS) produced by the mitochondrial electron transport chain. It also facilitates the generation of NADPH to reducing equivalents for ROS-protective pathways [17]. Except for plentiful ATP synthesis, Pentose phosphate pathway (PPP) is enhanced in the aerobic glycolysis. This pathway provides precursors for lipid and nucleic acid synthesis, which favors cell division [18]. Herein, the metabolic reprogramming can benefit both bioenergetics and biosynthesis, inhibit cellular apoptosis, and generate signal metabolites in favor of carcinoma cell growth.

Since the rate of aerobic glycolysis in carcinoma cells is so high that the speed of lactate production from glucose is approximately 10–100 times faster than the speed of complete oxidation of glucose in the mitochondria [11], it not surprising to observe that the concentration of lactate in the tumor tissues is 100 times as much as the blood [10]. It is estimated that the lactate concentrations range from 5 to 20 mM in the tumor microenvironment [19] and range from 10 to 40 mM in tumors [20]. Here, some questions arise: Is the excess generation of lactate a superfluous metabolic waste in carcinoma cells? If not, what is the pathophysiological function in carcinoma cells? As known, in mammals, lactate possesses two isomers: L- and D-lactate. Of what significance are they in carcinoma cells? In the following section, the content will involve L- and D-lactate production and metabolism in aerobic glycolysis, the research of lactate on cancer progress, hallmarks of cancer associated with the lactate, lactate related molecular signaling to better understand the role of lactate in cancer.

### 2.2. Metabolism of Lactate Isomers and Aerobic Glycolysis

Most tumor cells can reprogram metabolic procedures associated with increased levels of glycolytic enzymes and intermediates to enhance the glycolysis pathway [21,22]. Lactate is one of the well-known end-products of glycolysis. It is the simplest hydroxyl carboxylic acid and exists as 2 stereoisomers due to the chiral center at C2 [23]. Knowledge of the L- and D-lactate production in the Warburg effect will help us further understand the representative hallmarks in cancer progress and seek for the accurate anticancer targets.

#### 2.2.1. L-Lactate Production in Aerobic Glycolysis

Hexokinase (HK) is the first enzyme involved in glycolysis, catalyzing glucose into glucose 6-phosphate (G6P) [24]. G6P dehydrogenase (G6PD) irreversibly converts partially G6P to 6-phosphgluconate which is also known as the PPP [25]. In tumorigenesis, the utilization of PPP is frequently elevated [13]. In this step, G6P becomes oxidized to generate NADPH and ribose-5-phosphate (R5P)—a structural component of nucleotides. These transketolase reactions in the PPP convert glucose to ribose for nucleic acid synthesis, as well as generates NADPH, a reducing agent needed for synthesis reactions in tumor cells [11]. Yet, the P53 protein is reported to involve the “glycolytic stress response” by sensing an increased NADH: NAD^+^ ratio in highly glycolytic cells [17] and inhibit PPP by binding to G6PD [13,26]. In parallel to this process, G6P isomerase (GPI) catalyzes G6P to fructose-6-phosphate (F6P) in glycolysis [27]. Then, phosphofructokinase-1 (PFK1) catalyzes the rate-limiting phosphorylation of F6P to fructose-1,6-bisphosphate (FBP) [28]. FBP is cleaved into glyceraldehyde 3-phosphate (G3P) and dihydroxyacetone phosphate (DHAP) catalyzed by aldolase B [29]. G3P-dehydrogenase (GAPDH) can remove hydrogen from G3P to an NAD^+^ molecule for producing NADH or add a phosphate group to the G3P for producing 1,3-bisphosphoglycerate (1,3-BPG). Then, phosphoglycerate kinase (PGK) catalyzes 1,3-BPG and ADP to produce 3-phosphoglycerate (3-PG) and two ATP molecules. Phosphoglycerate mutase 1 (PGAM1), following, catalyzes the conversion of 3-PG to 2-phosphoglycerate (2-PG) [30]. After that, enolase catalyzes the dehydration of 2-PG into phosphoenolpyruvate (PEP) [31,32]. Finally, as one of the main PEP-consuming reactions, pyruvate synthesis is catalyzed by pyruvate kinase (PYK) [33]. In carcinoma cells, lactate dehydrogenase isoform A (LDHA) preferentially converts synthetic pyruvate to L-lactate by removing hydrogen from the NADH molecule in the final step of the glycolytic pathway [34], thereby regenerating NAD^+^ to maintain glycolysis [35,36], which serves as a substrate for GAPDH [37]. This is why the decreased GAPDH inhibits glycolysis [38,39], and the accumulation of L-lactate in carcinoma cells implies an increased intracellular NADH: NAD^+^ ratio [36] (Figure 1).

#### 2.2.2. D-Lactate Production in Aerobic Glycolysis

D-lactate, as an isomer of L-lactate, shares the same mass but has much lower amounts compared with L-lactate in mammals [40]. It is considered the “physiological inertia” in the body [41] due to the absence of metabolizing enzymes [42,43]. Previously, D-lactate is proved to be an important component of the cell wall of a lactic acid bacterium. Besides, bulk D-lactate can be detected in humans and ruminants in the rare metabolic condition of D-lactic acidosis [23]. For the past few years, D-lactate has also reported generation during aerobic glycolysis through the glyoxalase system [44], which is comprised of two enzymes, glyoxalase 1 (GLO1) and glyoxalase 2 (GLO2), and a catalytic amount of reduced glutathione (GSH) as a cofactor [45]. This system converts the metabolic intermediary product—methylglyoxal (MGO) [46] into D-lactate or GSH [44]. In the glycolytic pathway, MGO is a highly reactive three-carbon glycating metabolite [47] that mainly originates from triosephosphates (DHAP and G3P) para-metabolically and para-enzymatically when glucose is degraded [48,49,50]. Glyoxalases are involved in the detoxification of reactive MGO into D-lactate in a two-step reaction using GSH as a cofactor [48,51]. GLO1 (also named S-D-lactoylglutathione lyase) exists in humans, mice, yeast, and elegans [51]. It condensates MGO and reduces GSH to form S-lactoylglutathione [52]. Then, GLO2 hydrolyzes the S-lactoylglutathione and thereby, releasing D-lactate and regenerating GSH [48,52]. In breast carcinoma cells, astrocytoma, and prostate carcinoma cells, the levels of D-lactate are observed as elevated [48,53]. Furthermore, a recent study has demonstrated that produced D-lactate by lung carcinoma cells can shuttle into normal cells to lead to cancer-associated metabolic behavior, implying the role of elevated D-lactate concentration as a hallmark of cancer malignant metabolism [40] (Figure 1).

## 3. Current Advances of Lactate in Cancer

### 3.1. Breast Cancer

Breast cancer is the most frequently diagnosed cancer in women and ranks second among causes of cancer-related mortality in females worldwide [54]. The 5-year survival rate is 89% in females with primary breast cancer and less than 5% in patients with metastatic breast cancer [55]. The clinical hallmarks of breast cancer are stromal invasion and metastasis to regional lymph nodes or distant organs [56]. Bone, lung, liver, and brain are generally accepted as the primary target sites of breast cancer metastasis [57]. A previous clinical study has claimed that the lactate concentration is observed depending on the degree of progression of breast tumor tissue. For instance, the lactate concentration is 5.5 ± 2.4 mM in grade II and 7.7 ± 2.9 mM in grade III [58]. Similar to this result, the concentration of L-lactate in malignant breast tumor tissue is higher than in the benign counterparts [59], and tumor lactate in patients with triple negative breast cancer (TNBC) far exceeds that found in circulating blood [60]. The low perfusion or monocarboxylate transporters (MCTs) activity—MCT1 and MCT4 [60] in TNBC, may be the major cause of lactate accumulation in breast tumors and thereby, creates a local tumor microenvironment enriched in lactate produced by aerobic glycolysis [60]. Furthermore, Becker et al. found that L-lactate, produced by cancer-associated fibroblasts (CAFs), was delivered into breast carcinoma cells as fuel for growth and is dependent on the transport of MCT1 [61]. Distinguishment from the common breast cancer, TNBC lacks expression of an estrogen receptor (ER), progesterone receptor (PR), and human epidermal growth factor receptor 2 (HER2) [62]. It is interesting to investigate whether the expression of MCTs is affected by these receptors to influence the lactate shuttle between carcinoma and stroma cells in the tumor microenvironment and thereby, determining the cancer subtypes.

In breast carcinoma cells, the accumulation of lactate can promote the adhesion, migration, and invasion of carcinoma cells by serving as the signal modulator [63]. Lactate receptor—G-protein-coupled receptor 81 (GPR81), expression is observed as a high expression [64,65,66]. A further study demonstrates that GPR81 expression is conducive to multiple malignant phenotypes of carcinoma cells [64], implying the lactate-receptor signal is a potential therapeutic target for breast cancer. In parallel to GPR81, G protein-coupled receptor 132 (GPR132) can also serve as the macrophage sensor of the rising lactate in the acidic breast tumor milieu to promote the alternatively activated macrophage M2-like phenotype, which, in turn, facilitates cancer cell adhesion, migration, and invasion [67]. The M2-like phenotype also can be driven by lactate via the extracellular signaling-regulated kinase (ERK)/STAT3 signaling pathway [67]. Apart from the above molecular signals, 5 mM L-lactate is sufficient to induce the hypoxia induced factor-1 alpha (HIF-1α) expression to promote tumor-associated macrophages (TAMs) via overexpressing the HIF-1α-stabilizing long noncoding RNA [68]. The TAMs further enhance aerobic glycolysis [69] and inhibit apoptosis of breast carcinoma cells [68]. The inter-linked and mutually-reinforcing interaction of L-lactate and macrophages aggravate breast tumor progression. With regard to the role of D-lactate in breast cancer, to our knowledge, few related studies have been investigated. Considering that lactate comprises two isomers—L-lactate and D-lactate, the future research on breast cancer remains to distinguish the biological effect of two types of lactates, especially D-lactate production in glycolysis. Revealing the breast tumor-associated L- and D-lactate production, and their relation with respect to the phenotype of cancer, will provide a better understanding of the whole tumor progression.

### 3.2. Cervical Cancer

Cervical cancer is the fourth most common malignancy and the disease results in over 300,000 deaths annually worldwide [70]. Recent research has disclosed that, compared to healthy people, the plasmatic lactate concentration is significantly higher in patients with low- and high-grade cervical lesions and cervical cancer [71]. In cervical carcinoma cell lines, the secreted lactate concentration ranges from 1.5 to 3.8 mM after a 24 h period of incubation [71]. Inhibition of lactate synthesis or transport tends to decrease M2 markers of macrophage in the co-cultivated with human papillomavirus (HPV) positive cervical carcinoma cells and macrophages; as a result, the increase the T lymphocyte activation potential in the carcinoma cell lines [71] suggests that lactate inhibition may be a useful tool in anticancer therapies associated with immunomodulatory effects.

Human vaginal secretions have been reported to contain approximately 10–50 mM lactate through bacteria ferment and epithelial cells, of which D-lactate accounts for half of the total lactate [72]. There is no doubt that lactate isomers may play a potential role in the pathological mechanism of cervical cancer. Wagner et al. found that both L- and D-lactate can protect cervical carcinoma cell survival from chemotherapeutic treatment by inhibiting the activity of histone deacetylases (HDACs). The inhibited HDAC activity is beneficial to a more relaxed, transcriptionally permissive chromatin conformation and reduces the DNA damage response (DDR) by modulating the activity of key proteins such as an increased DNA-dependent protein kinase catalytic subunit (DNA-PKcs) [73]. In addition to epigenetic modification, lactate can also activate the GPR81 receptor signal pathway to achieve the survival of carcinoma cells by DNA repair, which is coordinated by MCTs transport [74,75,76]. The notable phenomenon observed by Wagner and his colleagues was that L-lactate primarily inhibited the cAMP accumulation while D-lactate strongly stimulated ERK phosphorylation, which was mainly induced by PKC [73], implying the disparate intrinsic activity of lactate isomers towards the GPR81 receptor signal transduction pathways. Based on the previous studies, Wagner et al. also considered the relationship between drug resistance depending on PKC activity and carcinoma cell survival. Their results suggested that the activated GPR81, stimulated by L- and D-lactate, up-regulated the protein and mRNA expressions of the ATP-binding cassette subfamily B member 1 (ABCB1) to enhance the doxorubicin resistance in the cervical cancer cell [74]. On the contrary, results of L-lactate favoring the progression of cervical cancer, Da et al. declared that the physiological concentration of L-lactate (10 mM and 20 mM) enhanced the phosphorylation of P38 to promote apoptosis in HeLa cells [77]. Wagner et al. declared that both L- and D-lactate (10 mM and 20 mM) may enhance the nuclear localization of DNA-PKcs to suppress retroviral transduction in cervical carcinoma cells [75]. Several factors may attribute to the paradoxical results: Different strains of cells react differently to lactate; for instance, DNA-PKcs-proficient cells among cervical cancer cells are less susceptible to lactate modulation. HeLa and CaSki cells respond to both lactate isomers, while C33A cells respond only to L-lactate [75]; lactate as the signal modulation regulates downstream multiple signal transduction related to cancers; lactate effects may be related to its volume in cancers in a link to the above research. Last but not least, the existence of pH caused acidification in the carcinoma cells and/or tumor microenvironment may affect the modulation of lactate-related signalings [71,77].

### 3.3. Lung Cancer

Lung cancer is the second most commonly diagnosed cancer after prostate cancer in men and breast cancer in women [78,79]. In North America and other developed countries, it is the leading cause of cancer-related deaths because of the difficulty for diagnosis in the early stage [80]. Higher lactate/3-PG labeling ratios have been noticed in patients with stage I and II lung cancers when they are observed at the time of the original clinical observation. In some cases, years before recurrence or metastases, the primary tumor is even observed with higher lactate/3-PG labeling ratios [20], implying that high lactate is more likely for the progress of lung cancer. In lung cancer model mice, the circulatory turnover flux of lactate exceeds that of glucose by approximately twofold and contributes to the tumor TCA cycle [3], suggesting that lactate can serve as the energy substrate for lung carcinoma cell growth.

Nonsmall cell lung cancer (NSCLC) is the main histologic subtype (85%) of lung cancer [81]. Surgical resections from patients with NSCLC show glucose metabolism-contrasting homeostasis after infusion of 13C-glucose, leading to considerably high levels of lactate [82]. Similarly, in the NSCLC mouse model, the contribution of lactate to the TCA cycle exceeds that of glucose [20]. In lung adenocarcinoma cell lines, upregulated gene expression of TMPRSS11B can enhance the lactate export to promote tumorigenesis [83]. The increased acidic environment along with lactate production promotes the formation of a snail/transcriptional coactivator with PDZ-binding motif (TAZ)/AP-1 complex and contributes to adaptive resistance in NSCLC in the end with the poor prognosis in advanced lung cancer [84]. Recent evidence has identified that lactate, as a characteristic of many NSCLCs, is exploitable for therapeutic targeting and manipulation to reprogram the TME and promote an oncolytic immune response [85]. For example, lactate can bind to its receptor GPR81 to induce the activation of PD-L1 which leads to the reduction of interferon-γ in lung tumor cells and apoptosis of co-cultured Jurkat T-cell leukemia cells for the evading host immunity [86]. Furthermore, 83% of tumor-bearing mice developed lung cancer and showed shorter survival when they were inoculated with the dendritic cells (DCs) treated with lactate. The results suggested that lactate caused the loss of DCs function to weaken the immune surveillance with reduced effector CD8^+^ T cells [87]. Besides, L-lactate is reported to subtly affect the transcriptome of the pro-inflammatory major histocompatibility complex (MHC)-II^lo^ TAMs to favor the typical M2 genes expression such as Cd163, Stab1, Lyve1, Tmem26, Folr2, Mmp9, Clec10a, Il4Ra, and Itgb3, that leads to the enhanced T cell suppressive capacity of these TAMs [88]. Of interest, the incubation of MHC-II^lo^ TAMs with L-lactate showed slightly elevated oxidative phosphorylation (OXPHOS) and enhanced glycolytic capacity, and glycolytic reserve. While in MHC-II^hi^ TAMs, L-lactate further reduces the ability of OXPHOS [88]. Hence, L-lactate may have different effects on mitochondrial metabolic regulation on the distinct macrophage phenotype in the carcinoma cells.

There are several problems to be solved here: What is the relationship between mitochondria and cancer immune escape? What is the effect of D-lactate on the mitochondria and immunosuppression of lung carcinoma cells? As for the research on D-lactate in lung cancer, Li et al. found that the D-lactate secreted by carcinoma cells can deteriorate the metabolic phenotype of cancer through the co-culture of the carcinoma and normal cells [40]. However, little research has focused on and revealed the molecular mechanisms of D-lactate in regulating lung cancer so far. Except for the immune response, lactate also participates in the mitochondria-related signals in NSCLC [89]. Dynamin-related protein (DRP1), as the regulator of mitochondrial fission, is reported to boost lactate utilization by reducing the production of reactive oxygen species (ROS) and protecting the carcinoma cells from oxidative damage [89]. However, in previous studies, L-lactate treatment can promote modest ROS production to activate PGC-1α mitochondrial biogenesis and NF-E2-related factor 2 (NRF2)—mediated antioxidant and excitotoxic signal transduction in SH-SY5Y [90] and L6 cells [91]. The contrary results may be due to the lactate isomers or the types of cell lines. If a certain proportion of L-and D-lactate treatment indeed has an effect on the tendency of the oxidative stress situation, the ratio of L-lactate/D-lactate may lead to the opposite fate of carcinoma cells. In this case, underlining the subtle metabolic changes of lactate in cancer cells and their TME may be a new direction for cancer treatment.

### 3.4. Pancreatic Cancer

Pancreatic cancer is the fourth leading cause of cancer death in the USA [92]. The incidence of this type of cancer continually rises with the lowest 5-year survival rate of 9% [79,93], and 95% of pancreatic cancer is classified as pancreatic ductal adenocarcinoma (PDAC). In the mouse model of pancreatic cancer, the activities of glycolytic metabolic-related enzymes (HK, PGK, pyruvate dehydrogenase kinase (PDK1), and LDHA) and the lactate transporter of MCT4 are far higher in the pancreatic tumor than the normal tissue [94], implying the potential role of lactate in tumor pathology. Under the hypoxic condition, in addition to the up-regulated enzymes and transporter, the pancreatic carcinoma cells can consume and release twofold more lactate than the normoxic cells after 48 and 72 h, implying that the pancreatic carcinoma cells possess a high glycolytic rate to produce and extrude lactate into extracellular space for the survival of carcinoma cells, guaranteeing their excellent aggressiveness [94]. For example, the lactate secreted by the PDAC cells can be uptaken by the mesenchymal stem cells as the energy substrate source of the pyruvate, which facilitates the de novo differentiation of mesenchymal stem cells into CAFs for tumor invasion and metastases [95]. Restraining the lactate metabolism by inhibiting the glycolysis or shuttle is reported to prevent tumor growth [96], as well as interfere with the expression of the lactate receptor GPR81 [97]. However, to our knowledge, little research has focused on the vital role of L- and D-lactate in the development of pancreatic cancer.

### 3.5. Prostate Cancer

Prostate cancer is a leading cause of cancer death among males following lung cancer worldwide [98,99]. Ippolito and his colleagues demonstrated that CAF-derived lactate can reprogram the lipid metabolism in prostate carcinoma cells for growth and metastasis [100]. Recent evidence has demonstrated that the lactate shuttle appeared to be linked to biochemical recurrence after surgery in prostate cancer patients, suggesting that lactate and its metabolism were potentially useful poor prognostic markers [101,102,103]. Fiaschi et al. have found that the prostate cancer cells underwent metabolic reprogramming to support the growth of carcinoma cells that gradually tended to depend on lactate-derived anabolic metabolism by increasing the expression of MCT1 and MCT4 [102]. Ippolito et al. have demonstrated that CAF-derived lactate can promote prostate carcinoma invasion which was dependent on the regulation of MCT1 and LDHB. The intracellular lactate herein induces the HIF-1α stabilization and SIRT1-PGC-1α signaling pathway to enhance the mitochondrial metabolism by altering the NAD^+^/NADH ratio [104]. Except for involvement in the mitochondrial metabolism via signal mediation, lactate can also work as the direct fuel for mitochondria in the prostate. Bari’s team has revealed the role of L-and D-lactate in mitochondrial metabolism. They claimed that L-lactate can be uptaken by both prostate normal and carcinoma cells, and metabolized by their mitochondria. With a higher mLDH (mitochondrial L-lactate dehydrogenase) activity in carcinoma cells, it can be presumed that a higher volume of pyruvate and NADH production supports the energy demand for the pathological development of prostate cancer [105]. A subsequent study reported that D-lactate can also shuttle into the mitochondria as an energy substrate for malate production in the prostate normal and carcinoma cells. Interestingly, this malate efflux rate caused by D-lactate metabolism is twofold in the prostate carcinoma cells than the normal cells. The process of D-lactate can facilitate the elimination of MGO for ROS reduction, the production of NADPH, and the synthesis of fatty acids which is vital for the viability and proliferation of carcinoma cells [53]. Up to date, the lactate oxidative metabolism in the prostate mitochondria is based on the putative LDH located at the mitochondrial inner (an mLDH for L-lactate metabolism [91,105] and D-lactate dehydrogenase (LDHD) for D-lactate metabolism [53,106,107]); whether the phenomena occur in other cancers remains to be verified. As mentioned in the above context, lactate can influence receptor signaling, immune escape, and DNA repair in cancers. Getting the whole picture of how lactate metabolism shapes the development of prostate cancer may provide a comprehensive knowledge hierarchy and precise treatment strategy.

### 3.6. Liver Cancer

Liver cancer is an extraordinarily heterogeneous malignant disease among tumors [108], which is the fifth most frequent fatal malignancy worldwide and most patients survive less than a year [109]. Hepatocellular carcinoma accounts for 70–85% of total liver cancer and arises most frequently within the background of chronic liver disease [108]. Recent evidence has revealed that the increased lactate abundance in both plasma and liver tissues was highly associated with the occurrence of hepatocellular carcinoma [110]. The elevated lactate uptake can promote ATP production to supply energy for the growth of hepatocellular carcinoma cells [111]. In addition, the lactate can also be absorbed by Treg cells to promote the nuclear factor of activated T cells 1 (NFAT1) translocation into the nucleus for enhancing the expression of PD-1 in liver tumors and thereby, leading to immune escape [112]. Further supportive evidence for lactate facilitating the development of liver cancer is the application of a genetic tool for interfering the glycolysis. For example, inhibition of lactate production by knockdown of aldolase A (ALDOA) [113] or the HK [114] expression in the process of glycolysis can hamper cell proliferation, migration, and tumorigenesis in the hepatocellular carcinoma cells.

Recent studies have found that L-lactate treatment inhibited the phosphorylation of AMP-activated protein kinase (AMPK) to activate the sterol regulatory element-binding protein 1 (SREBP1) and its downstream stearoyl-coenzyme A (CoA) desaturase-1 (SCD1) in order to drive the ferroptosis resistance and protect the cell from death following the intracellular decreased ratio of AMP: ATP [111]. In addition, exogenous L-lactate treatment can also induce the N-myc downstream-regulated gene family member 3 (NDRG3)/Raf/ERK hypoxia signaling axis to stimulate the angiogenesis and tumor growth of hepatocellular carcinoma cells [115]. From what has been discussed above, interfering with key enzymes or genes of the glycolysis process or reducing L-lactate levels in the tumor microenvironment may exploit an efficient therapy against liver cancer.

## 4. Lactate Metabolism Related Prognostic Markers in Cancer

During the Warburg effect, the production of lactate (L- and D-lactate) remodels the micro-environment in favor of carcinoma cell growth [6]. It creates a tumoral acidic microenvironment [116,117] and thereby, promotes higher tumoral cell proliferation, survival, migration, invasion, and angiogenesis [8,117], and suppression of anticancer immune response [118]. Several molecular pathways work in concert toward the lactate metabolism in the TME, including lactate production and conversion (LDHA, LDHB, LDHD, GLO1, and GLO2), transport (MCT1 and MCT4), and receptor interaction (GPR81 and GPR132) (Figure 2). The expression levels of these molecules are often observed alterant, and associated with poorer prognoses in cancer. For this reason, they are promising prognostic biomarkers and valuable therapeutic targets for clinical cancer treatment. In this section, we will discuss the research advance of these molecules in various cancer types.

### 4.1. LDH

LDH is a tetrameric enzyme that belongs to the 2-hydroxy acid oxidoreductase family [119] and catalyzes the interconversion of pyruvate and lactate during the processes of glycolysis and gluconeogenesis [120,121]. The activation of oncogenic pathways often results in high serum LDH levels in various types of cancer such as ovarian cancer [122], cervical cancer [123], lung cancer [124,125], prostate cancer [126], and primary pancreatic lymphoma [127], which are associated with drug resistance [128]. Hence, it is no accident that an elevated level of LDH is believed to be a hallmark of aggressive cancers and a negative prognosis. LDH exists in two different subunits—LDHA and LDHB [129], which can be assembled into five different combinations in the way of homotetramers or heterotetramers: LDH1 (four LDHB subunits), LDH2 (three LDHB subunits and an LDHA subunit), LDH3 (two LDHB and LDHA subunits), LDH4 (an LDHB subunit and three LDHA subunits), and LDH5 (four LDHA subunits) [130,131]. Knockdown of LDHA or LDHB knockdown is reported to reduce the LDH activity and lactate production in breast carcinoma cells [132]. Recently, Ždralević and his colleagues have reported that glycolysis and lactate secretion cannot be completely contained in human colon adenocarcinoma and murine melanoma cells, but double blockage of the LDHA and LDHB for fully suppressing LDH activity [133]. Therefore, combined targeting of LDHA and LDHB will be more effective anti-glycolytic-based therapeutic strategies for cancer treatment.

The primary function of LDA is to convert pyruvate to L-lactate dependent on the oxidation of NADH to NAD^+^ [131] which is the predominant form in highly glycolytic cells [134], and a high LDH5 content in tumor cells is linked with an aggressive phenotype in colorectal adenocarcinomas [135]. A previous study has demonstrated that the LDHA gene promoter showed higher hypomethylate in the breast CAFs, suggesting that epigenetic modification may be one of the causes of the increased activity of LDHA during the progression of cancers [61]. In CD8^+^ T effector cells, the activity of LDHA increases in response to the phosphoinositide 3-kinase (PI3K) signal. In turn, LDHA deficiency will disturb cellular redox control and weaken ATP production to inhibit PI3K signaling [136]. As known, PI3K regulates the growth signal for carcinoma cell division. Therefore, LDHA may serve as the switch of the Warburg effect and affect the multiple oncogenic signaling. Inhibition of LDHA by genic tools or pharmacological reagents has been reported to induce oxidative stress [137,138,139], decrease cellular proliferation [140], promote cell death [138], activate apoptosis [139,141], enhance tumor suppressor p53 expression [17], suppress the inflammatory response [129], and restore the immune functions [129,142] in a variety of cancer cell lines. The current pharmacological LDHA antagonist includes oxamate [17,141], GSK2837808A [140,142,143], and FX11 [138,144]. Additionally, overexpression of miR-200c [145] and miR-34a [146] can also directly inhibit the activity of LDHA to suppress the proliferation and migration in the carcinoma cells.

As for LDHB, it mostly converts lactate into pyruvate and NADH [130]. In breast cancer, the expression of LDHB is lower in malignant tumors than the benign tumors and is preferential in cancer-associated adipocytes [59], implying the tissue specificity of LDHB expression in the tumor cells and stromal cells of TME. In the PDAC, LDHB protein is overexpressed in tumors and is associated with worse survival [147]. In colorectal cancer, Krüppel-like transcription factor 14 (KLF14) targets LDHB to inhibit glycolysis and is associated with higher overall survival and disease-free survival [148]. Epigenetic modification is reported to be vital for the activity of LDHB. For example, sirtuin 5 (SIRT5) can deacetylate LDHB at lysine 329 to accelerate the growth of colorectal cancer [149], and the increased phosphorylation of LDHB at serine 162 can promote NAD^+^ regeneration, glycolytic flux, lactate production, and glycolytic intermediate generation [150]. Additionally, miR-375 [151] and miR-335-5p [152] can also directly suppress the LDHB expression to inhibit the growth, proliferation, and migration in the carcinoma cells. On the contrary, in pancreatic cancer, suppressed expression of LDHB aggravates glycolysis to promote proliferation, invasion, and migration [153]. Furthermore, in liver cancer, decreased expression of LDHB is reported to induce mitochondrial defects and thereby, carcinoma cell invasiveness [154]. In breast cancer, LDHB down-regulation induced by the tumor-derived miR-375 in the TAMs will drive macrophage polarization and subsequent tumor growth [155], whereas the decreased activity of LDHB in the carcinoma cell may inhibit tumor growth [156]. Hence, LDHA is a promising predictor of poor prognosis and a target for anticancer therapy, whereas the significance of LDHB in tumor development is still elusive. Recently, a small-molecule LDHB selective inhibitor named AXKO-0046 has been developed [157]. Application of this pharmacology antagonist and/or the genetic engineering technology may facilitate our knowledge of the LDHB function in cancer.

Except for LDHA and LDHB, LDHD also has been described in some cancers [120]. LDHD is a flavoenzyme [53] and responsible for D-lactate metabolism [158] which exists in lactobacillus strains [121], human tissues with a high metabolic rate [159], and mitochondria [53,159]. In a cohort study of renal cell carcinoma, the LDHD expression in the tumor is reported to be influenced by the tumor’s pathological T stage, and the down-regulated LDHD is associated with poor overall survival [120]. In uterine sarcoma, the expression of LDHD is far higher than that in patients with uterine myoma or cellular leiomyoma, suggesting the possibility of LDHD for aiding in the pathological diagnosis of tumor types [160]. As D-lactate is considered to be released from carcinoma cells and its role in cancer has been gradually uncovered, combined with the detection of D-lactate and LDHD in blood and/or tissues, it will be a potential predictive marker of diagnosis of cancers. Alpha-hydroxy acid 2-hydroxy-3-butynoate (αHB) is reported to be the inactivator of the LDHD [161]. Nevertheless, this compound has not been applied in tumor animal models or carcinoma cell lines with no recognition of its drug sensitivity and toxicological response. A future study applying αHB may be an alternative choice for anticancer drug therapy and an optional inhibitor for revealing the characteristics and biological functions of LDHD in different types of cancer.

### 4.2. Glyoxalases

GLO1 and GLO2 belong to the glyoxalases, which are the key metalloenzymes in the glycolytic pathway, that involve the detoxification of reactive methylglyoxal into D-lactate by using GSH as a cofactor [45,48,162,163]. Among glyoxalases, GLO1 is an active detoxification enzyme in both cancerous and normal cells [74]. In established human tumors, the increased expression and activity of GLO1 are an oncogene that is associated with tumor growth [164]. In the NSCLC mice, the expression of GLO1 is required for the growth of tumors [23]. Herein, overexpression of GLO1 is permissive for carcinoma cells with high glycolytic activity and is a cause of multi-drug resistance [164,165,166]. It is produced by the NRF2 pathway and GLO1 amplification [164,167]. A previous study has demonstrated that NRF2 was observed as up-regulated, accompanied by the aggravated malignant phenotype of cancers, such as liver [168], lung [169,170], and breast cancer [171]. The activated NRF2 is reported to induce the expression of GLO1 [172,173]. Additionally, GLO1 is the main amplified gene of locus 6p21.2 in human cancers, providing a potential target for therapy in cancers with GLO1 amplification [167]. In the human genome, increased GLO1 copy number and expression are found in tumors [174]. Twofold and higher amplification of GLO1 in tumor tissues is identified in the breast, sarcomas, NSCLC, bladder, renal, and gastric cancers [167]. In fact, knockdown of GLO1 can reduce the migration, invasion, colony formation, tubule formation, proliferation, and cell viability of carcinoma cells in breast cancer [175,176]. In colon cancer, the silence of GLO1 can also inhibit these tumor properties via up-regulating the transcription-1 (STAT1) expression and the B-cell lymphoma-2 (Bcl-2)/Bcl-2-associated X protein (Bax)—mediated apoptosis signal [177]. Except for the genic tool of inhibiting GLO1, current studies have exploited various pharmacological inhibitors and related prodrugs with the purpose of promoting their development toward clinical application [165,166,178], such as curcumin, luteolin, delphinidin, methyl-gerfelin, tropolone, 18-glycyrrhetinic acid, 6-sulfamoylsaccharin, zopolrestat, and S-p-bromobenzylglutathione cyclopentyl diester. Of interest in that regard, in the non-malignant state of liver cancer, GLO1 is a tumor suppressor gene [179]. In this case, it is necessary to explore the role of GLO1 and its inducers—trans-resveratrol-hesperetin (tRESHESP) in the clinical chemoprevention effect of cancers [164,180].

Although the role of GLO2 is little investigated, it is also reported to be involved in the process of cancers in recent research [181,182]. In prostate cancer, GLO2 is positively associated with the malignant phenotype [181]. A further study has revealed that GLO2 can inhibit the expression of p53 to stimulate proliferation and elude apoptosis for tumor growth [182]. This evidence represents the potential of GLO2 as a diagnostic and prognostic indicator for prostate cancer. Future studies need to investigate the relationship between GLO2 and other cancers. Generally, improving the understanding of glyoxalases in cancer pathogenesis will help assess the importance of the further source of biomarkers for tumor prognosis associated with the D-lactate metabolism, especially since GLO1 is linked to multidrug resistance in cancer chemotherapy.

### 4.3. MCTs

MCTs belong to the SLC16 gene family which is encoded by SLC16A1, SLC16A3, SLC16A7, and SLC16A8 [183]. They transport the proton-linked monocarboxylate metabolites such as pyruvate, lactate, and ketone body [184,185,186]. The high expression of MCTs is extensively characterized in multiple cancer cell lines and tumor types [20,187,188,189,190]. In cancer, H^+^-coupled transport by MCTs tends to drive lactate from the interstitium into tumor cells to maintain the concentration gradients of lactate and Ph [191,192], and they mold a phenomenon called “metabolic symbiosis” between hypoxic and aerobic carcinoma cells, where lactate secreted by glycolytic cancer cells is exported by MCT4 and transported into oxidative cancer cells by MCT1 as an oxidative fuel [188,193,194]. Besides working as the substrate for carcinoma cells, lactate also exerts the modulator effect for signal transduction in the metabolism of endothelial cells [194,195,196]. For example, lactate can stimulate the nuclear factor kappa B (NF-κB)/ Interleukin-8 (IL-8) or the HIF-1α signaling pathway to promote tumor angiogenesis and growth when it is released from tumor cells through MCT4 and imported into endothelial cells via the MCT1 [197,198]. Because of the important metabolic roles of MCTs (especially MCT1 and MCT4) in tumor cells, it is considered the prognosis of cancers and developed into the targets for anticancer therapy drugs. In humans, high MCT1 and MCT4 expressions are usually associated with poor prognosis [187], whereas MCT2 expression correlates with a favorable outcome [199]. Since MCT1 and MCT4 are deemed as the predominantly expressed isoforms in cancer [194], in this context, MCT1 and MCT4 have been proposed as potential anticancer therapeutic targets in cancers. To our current knowledge, the targeted drug development of MCT1 inhibitors has been in the advanced development phase, while MCT4 inhibitors are still in the discovery phase [187].

Fiaschi et al. found that inhibition of MCT1 by α-cyano-4-hydroxycinnamate (CHC) or siRNA interference inevitably decreased the tumor volume and thereby restrained prostate carcinoma cell survival [102]. Combining CPI-613 (as known as Devimistat which is a potent inhibitor for TCA enzymes) with CHC can inhibit pancreatic carcinoma cell proliferation and induce apoptosis [96], suggesting the potential for combined use of MCT1 inhibitor and other anticancer drugs for cancer therapy. Additionally, the application of the MCT1 inhibitor-AZD3965 can elevate the lipid ROS levels by 52.8% and hence, ferroptosis in the liver tumor tissue, which can repress the tumor growth and prolong the average survival time of mice more than 1 month [111]. Interestingly, the chronic pharmacologic blockade of MCT1 by CHC can decrease the tumor cell oxygen consumption and delay the tumor growth in the mouse models of lung cancer and the human colorectal adenocarcinoma cell line [188], whereas the antitumor efficacy is restricted to expressing MCT1 located at the plasma membrane of carcinoma cells [188]. This implies that the function of MCT1 in importing lactate into aerobic carcinoma cells is important for the survival of carcinoma cells. Unlike the epithelial-derived malignant cancers, the function of MCT1 seems to guarantee the lactate efflux from lymphoma cells protein in hematological cancers which a lack of MCT4 protein. In diffused large B-cell lymphoma (DLBCL) and Burkitt lymphoma (BL) cell lines, inhibition of MCT1 by the antagonists such as AZD3965, AR-C122982 (as known as SR13800), and AR-C155858 (as known as SR13801), can intercept lactate efflux for the sake of intracellular acidification and thereby, delay tumor lymphoma growth [200,201,202,203]. In addition, MCT1-targeted drugs may also have an effect on immunosuppression in hematological cancers. For example, inhibition of MCT1 during jurkat-T cell activation can prevent the proliferation of T cells [204]. In the Raji lymphoma mice model, blockage of the activity of the MCT1 by AZD3965 can boost the abundance of DCs and mature natural killer (NK) cells in the tumor tissue to improve immune cell infiltration [205].

In regard to the role of MCT4 in cancers, Wang et al. demonstrated that CD147-K234me2 can promote MCT4 translocation from the cytoplasm to the plasma membrane to enhance lactate export and thereby, lead to exacerbated progression and shortened overall survival of NSCLC [206], suggesting that the abundant expression of MCT4 is highly associated with the poor prognosis of cancer. Hypoxia-induced HIF-1α can also stimulate the MCT4 promoter and increase its expression in the carcinoma cell lines [207]. Therefore, the antitumor drugs that target silencing the CD147-K234me2 or HIF-1α are a potential choice. Recent evidence has shown that the overexpression of MCT4 in CAF can be inhibited by the anti-oxidant N-acetyl-cysteine (NAC) during breast cancers, suggesting that targets on ameliorating oxidative stress and regulating the MCT4 expression may control the pathological progress of such cancers [189]. Notably, on the contrary, simply increased MCT4 expression for lactate extrusion in fibroblasts results in the death of tumor stroma when the TME acidification rises and lactate is incapable of intaking into epithelial cancer cells [189]. The results indicate a prominent energy transfer mechanism of lactate shuttling from hypoxic to aerobic carcinoma cells during malignancy. Generally, further in-depth studies are needed to underline the metabolic symbiosis within tumor cells or between tumor cells and stromal cells (fibroblasts, endothelial cells, and immune cells) which will be conducive to understanding the role of lactate transport and metabolism in various cancers, and offer opportunities for developing a new rationale and effective strategy for clinical anticancer therapy. We speculate a novel limotherapy for treating cancers that disturbs the relationship of metabolic symbiosis by controlling the MCT4 expression and/or specifically blocking the MCT1 expression, the possible mechanisms including lactate shuttles, immune suppression, oxidative stress, and neovascularization.

### 4.4. GPR81

GPR81 (also named HCA1) is identified as the G_i_ type G protein [208,209], which is the only known physiological endogenous receptor of lactate [210,211]. Recent studies have identified GPR81 in several carcinoma cell types, including colon, breast, lung, hepatocellular, salivary gland, cervical, and pancreatic carcinoma [64,65,66,97], which functions as a tumor promoter by sensing extracellular lactate concentration [212]. In breast cancer, GPR81 can enhance carcinoma cell proliferation, promote migration and invasion, boost angiogenesis, and inhibit apoptosis [64,66]. A previous study has declaimed that GPR81 is highly expressed in breast cancer cell lines but not in normal breast epithelial cells. In fact, the survival of breast cancer cell lines (BT-474 and HCC1954) depends on the GPR81 mediation [65]. Knockdown of GPR81 can decrease the lactate release from carcinoma cells. Under the circumstances, glycolysis is impaired against the ATP production for tumor growth in breast cancer cells [66]. In addition to obstructing lactate metabolism, the knockdown of GPR81 also affects the PI3K/protein kinase B (AKT) signal pathway. Blocking the GPR81 activation will abrogate the PI3K/AKT downstream cAMP response element binding protein (CREB), which induces the production of the pro-angiogenic mediator amphiregulin (AREG) and thereby the angiogenic effect [64]. In cervical cancer, the activated GPR81 can enhance the expression levels of DNA repair proteins for improving the efficiency of DNA repair, including breast cancer type 1 susceptibility protein (BRCA1), Nijmegen breakage syndrome 1 (NBS1), and DNA-PKcs [76]. Further in-depth research also demonstrates that the GPR81-PKC signal up-regulated the expression of ABCBI for promoting doxorubicin resistance and carcinoma cell survival [74]. In lung cancer, the GPR81 expression level is observed higher in lung cancer tissues than in the adjacent noncancerous lung tissues, indicating the important role of GPR81 in the pathogenesis of lung cancer [213]. On the one hand, under the condition of lactate persistent stimulation, the Snail is activated to enhance the STAT3 activity and then binds to the GPR81 promoter for up-regulating its expression for the effect of carcinogenesis [213]. On the other hand, GPR81, as a known G_i_ protein [214], can inhibit protein kinase A (PKA) activity via reducing intracellular cAMP levels. In this regard, the activity of TAZ was enhanced and then it interacts with the transcriptional enhanced associate domain (TEAD) for the induction of PD-L1 [86]. In pancreatic cancer, the GPR81 expression level is correlated with the rate of tumor growth and metastasis. Silence of GPR81 results in the reduction of mitochondrial activity by approximately 50% [97].

As emerging evidence shows that mitochondria and their consequent OXPHOS are essential in the development and diagnosis of cancers in the terms of initiation, metastatic potential, progression, and drug resistance of cancers [14,215,216]. Shifting lights on investigating causal mechanisms in further in-depth research will offer a new perspective on tumor metabolism where the lactate-GPR81 signal regulates the mitochondrial function in cancers. In liver cancer, the GPR81 expression level drastically increases in carcinoma tissues and is in connection with poor treatment outcomes and terrible prognosis [111], which is vital for the growth, survival, and immune evasion of carcinoma cells. Knockdown of GPR81 induces higher lipid ROS levels and leads to the ferroptosis effect of cell death [111]. GPR81 can also cooperate with MCT to regulate lactate metabolism in cancer.

Therefore, the result of GPR81 activation in the regulation of the cancer progress is multipath, including angiogenesis, DNA repair, chemoresistance, immune evasion, mitochondrial metabolism, and oxidative stress. Besides, GPR81 can also mediate the levels of lactate transporters. For example, GPR81 knockdown leads to the down-regulation of MCT1 by 85% [111]. This means that GPR81 can also cooperate with MCT to influence lactate uptake in tumors and TME, thereby the metabolic wiring in cancer. Applying the genic tool or chemotherapeutic agent that reveals the complicated mechanism of GPR81 in the pathological progression of cancer and the potential as a drug target, will be conducive to the development of cancer treatment clinical strategies. Unfortunately, although several agonists for GPR81 have been developed in the marketplace [217], a specific antagonist of this receptor is still in a gap.

### 4.5. GPR132

GPR132 (also named G2A) is a heptahelical cell surface receptor that activates the RhoA expression and induces the phenotypes characteristic of oncogenic transformation [218]. High expression of this receptor in macrophages [219,220] will confer TAMs, the tumor-promoting effects in terms of inflammation and tumor progression promotion [219]. As GPR132 is identified as a novel acidic extracellular pH sensor [221], it can be activated following the rising lactate production [67] from the Warburg effect and thereby, mediating some tumor effects under the acidic TME. In fact, the tumor-secreted lactate can accelerate cell adhesion, migration, and invasion in breast cancer by facilitating the macrophage M2 phenotype, which is dependent on the activation of a GPR132-dependent manner [67] (Figure 3). Additionally, growing evidence demonstrates that the activation of G2A may contribute to the recruitment of T cells to sites of inflammation [222,223], implying the immune regulatory function of this receptor. Considering the critical function of lactate modulating and shaping the immune cells [129,224,225], exploring the function of GPR132 not only conduces the understanding of the role of GPR132 in TME but also further reveals the elaborate carcinogenesis of lactate metabolism in cancer pathology. Current research has reported that lysophosphatidylcholine can function as an antagonist of GPR132 [226], and rosiglitazone can also be used as the pharmacological antagonist of this receptor via the activation of peroxisome proliferator-activated receptor gamma (PPARγ) [219]. Hence, biological agents can inhibit the expression of GPR132 in stromal cells of TME and be beneficial for uncovering the value of this GPR132 as a therapeutic target in clinical anticancer treatment.

## 5. The Downstream Molecular Signalings of Lactate (L-and D-Lactate) Mediation in Cancer

There is growing evidence for cross-talk between cancerogenic signaling pathways and lactate metabolic control in cancer. Lactate is deeded as the lactormone [227] that mediates intracellular molecules. In this part, we will introduce the downstream signalings mediated by lactate in cancer (Figure 3).

### 5.1. MAPK

Mitogen-activated protein kinase (MAPK) signaling is fundamental in cancer progression control which regulates proliferation, apoptosis, and immune escape [228]. ERK1/2 and p38 are members of the MAPK pathway [229]. Recent evidence shows that these two molecules are also involved in lactate-mediated carcinoma cell growth and survival. In breast cancer, tumor-derived lactate can activate the ERK1/2 and its downstream STAT3 signaling to induce macrophage M2 polarization for tumor growth and angiogenesis [230]. In cervical cancer, treatment of the HeLa cells with L-lactate can phosphorylate the p38 to subsequently stimulate apoptosis by up-regulating the expression of Bax and Caspase 3 and decreasing the expression of BCL-2 [77].

### 5.2. HIF-1α

The hypoxic response is an intrinsic feature of solid tumors [231] and the classical hallmarks of cancer [232]. The insufficient intratumoral oxygen supply is rooted in a chaotic, deficient tumor microcirculation [233]. In turn, a hypoxia microenvironment boosts cancer metabolic rewiring and development, such as metastasis [234], tumor growth [235], angiogenesis [236], and tumor immune response [237,238]. Therefore, hypoxia is always associated with aggressive cancer phenotypes and poor patient prognosis [231]. Hypoxia rewires the metabolism in the TME at the expense of switching oxidative phosphorylation to glycolysis [239]. HIF-1α is a transcription factor that acts as a regulator of oxygen homeostasis by binding to hypoxia response elements (HREs) and activating the transcription of hundreds of genes in response to reduced oxygen availability [240]. For instance, it can promote angiogenesis by stimulating the transcription of angiogenic cytokines and cell proliferation by mediating the G1 cell cycle arrest [239]. Meanwhile, it adapts carcinoma cells to hypoxic and nutrient-deprived conditions [231] via increasing glucose transport, glycolysis, and lactate [241,242].

Lactate production during the Warburg effect in turn can stimulate the expression of HIF-1α to aggravate the malignant phenotypes of cancer [243,244,245]. In breast cancer, L-lactate induces the HIT-1α to enhance aerobic glycolysis and promote the survival of carcinoma cells [68]. In prostate cancer, inhibiting lactate import into the carcinoma cells cripples the stabilization and activation of HIF-1α and subsequently impaired cell invasive skills [246]. A similar effect also exists in the oxidative carcinoma cell lines such as SiHa, HeLa, and FaDu [245]. Additionally, reduced ROS levels along with decreased lactate in CAFs down-regulates the HIF-1α accumulation. When the HIF-1α is blocked, it also negatively regulates lactate uptake into carcinoma cells by inhibiting both MCT1 and MCT4 expression [102], as a result, retaining the carcinoma cell growth. As the phenomenon of lactate promoting the intracellular ROS levels has been observed in other cell lines [90,91], reduced lactate production is likely to negatively influence the stabilization of HIF-1α in carcinoma cells via decreased ROS. Therefore, there is a visible interaction effect between lactate and HIT-1α in the TME.

### 5.3. NDRG3

Except for the classical HIF-1 mediated hypoxic response, NDRG3, identified as the hypoxia-inducible lactate sensor in 2015 by Yeom and his colleagues, also mediates a lactate-dependent signaling pathway in hypoxia [115,247]. It is negatively regulated by oxygen at the protein level via the proteasomal pathway [247]. In the later phase of hypoxia, the accumulated lactate can bind to the NDRG3 to induce the phosphorylation of C-Raf for the activation of downstream ERK1/2 signaling [247], which promotes angiogenesis and cell growth [248]. In the breast and cervical carcinoma cells, the expression of NDRG3 is increased as the oxygen concentration declines [115]. In hepatocellular carcinoma cells, the NDRG3 protein expression is highly correlated with the activity of angiogenesis, anti-apoptosis, and proliferation via analyzing the genomic activity [115]. Overexpression of NDRG3 is reported to highly induce the phosphorylation of C-Raf at Ser338 and ERK1/2. With the knockdown of the NDRG3 gene in hepatocellular carcinoma, the phosphorylation of C-Raf and B-Raf (at Ser445) is abrogated, and in consequence, the blocked angiogenesis and hypoxic cell growth [115]. Given the above, the target for this lactate/NDRG3 cascaded hypoxia signaling may be a novel molecular strategy of anticancer therapy.

### 5.4. PI3K/AKT

The PI3K/AKT signaling pathway is hyperactivated in various human cancer types [249,250,251] and is the onset or progression of cancers [252]. The activated AKT can trigger an enhanced glycolytic rate by up-regulating the HK2 activity for carcinogenesis [11,253], and during cancer, PI3K/AKT pathway enhances drug resistance and intercepts the anticancer therapy [254]. Therefore, the inhibition of PI3K can contribute to the restricted proliferation, suppressive growth, and increased death of carcinoma cells [255,256]. In breast cancer, the lactate receptor GPR81 can activate the CREB to promote angiogenesis by up-regulating PI3K/AKT [64]. This suggests that lactate may serve as a metabolic transmitter to mediate this pathway. Although several inhibitory drugs target this signaling perform efficacy against cancers, the therapeutic efficacy of them is unsatisfactory due to intrinsic and acquired resistance [254]. Therefore, understanding how lactate influences the PI3K/AKT axis will make for a novel idea for the inhibitory anticancer agent development targeting this molecule via redressing the tumor metabolism.

### 5.5. NF-κB

NF-κB is involved in the regulation of biological responses, including immune responses and inflammation, as well as in oncogenesis [257]. The hyperactivation of NF-κB is reported to enhance the aggressive skills of invasion [258] and migration [259] in carcinoma cells. Growing evidence has shown the close relationship between lactate regulation and the activity of NF-κB. In breast and colorectal cancer, lactate can activate the expression NF-κB and produce the IL-8 to promote the maturation of the tumor neovasculature via generating the ROS and phosphorylating the IκBα [197]. As the macrophage M2 phenotype is associated with less NF-κB expression, the limitation of the activity of NF-κB in TAMs promotes cancer progression [260,261]. In a recent study of cervical cancer, the lactate secreted by carcinoma cells is observed to down-regulate the expression of NF-κB and thereby, tend the TAMs into the M2 phenotype characteristics [71]. This indicates that lactate can regulate the NF-κB activity in a tissue-specific manner. Further research, that reveals how lactate precisely regulates the NF-κB signaling in different stromal cells and favors tumor growth, may be beneficial to the development of NF-κB targeted drugs.

### 5.6. Wnt Signaling

Wnt signaling is a highly conserved signaling pathway that plays a critical role in tumorigenesis in different organs, and affects the tumor cell and immune microenvironment [262,263]. The activation of Wnt contributes to tumor recurrence and has been observed in several cancer types, such as breast, colorectal, lung, endometrial, and hematologic [262,264,265,266]. A previous study has reported that the lactate/GPR81 signal can activate the Wnt to promote the proliferation of intestinal stem cells [267] and retinal angiogenesis [268]. However, to our knowledge, little research has focused on the lactate/GPR81/Wnt signal axis to date. Insights gained from understanding how the Wnt pathway involves in cancer cell maintenance and growth in link with the lactate metabolism may serve as a paradigm for deepening our molecular understanding of how lactate educates cancer progress, which provides a novel signaling target for anticancer therapy.

## 6. Conclusions

Malignant carcinoma cells often exhibit an increased dependence on high rates of aerobic glycolysis which is called the Warburg effect. Lactate production for carcinogenesis is the explanation and purpose of the Warburg Effect. Accordingly, therapies to limit lactate metabolism and downstream signaling molecules should be priorities for discovery. Therefore, excessive lactate levels are common in tumors and are closely related to the progression of cancer. L- and D-lactate are the isoforms of lactate. However, how they influence the carcinoma and stroma cells in the TME and perform the cancerogenic downstream signaling cascade is still obscure, especially the D-lactate. An in-depth understanding of the metabolic changes caused by L-and D-lactate in the tumor may lead to the development of novel anticancer strategies targeting multiple molecular pathways, including MAPK, HIF-1α, NDRG3, PI3K/AKT, NF-κB, and Wnt, which might improve the effectiveness and/or overcome chemoresistance of inventive drugs.

## Figures and Tables

**Figure 1 cancers-15-00087-f001:**
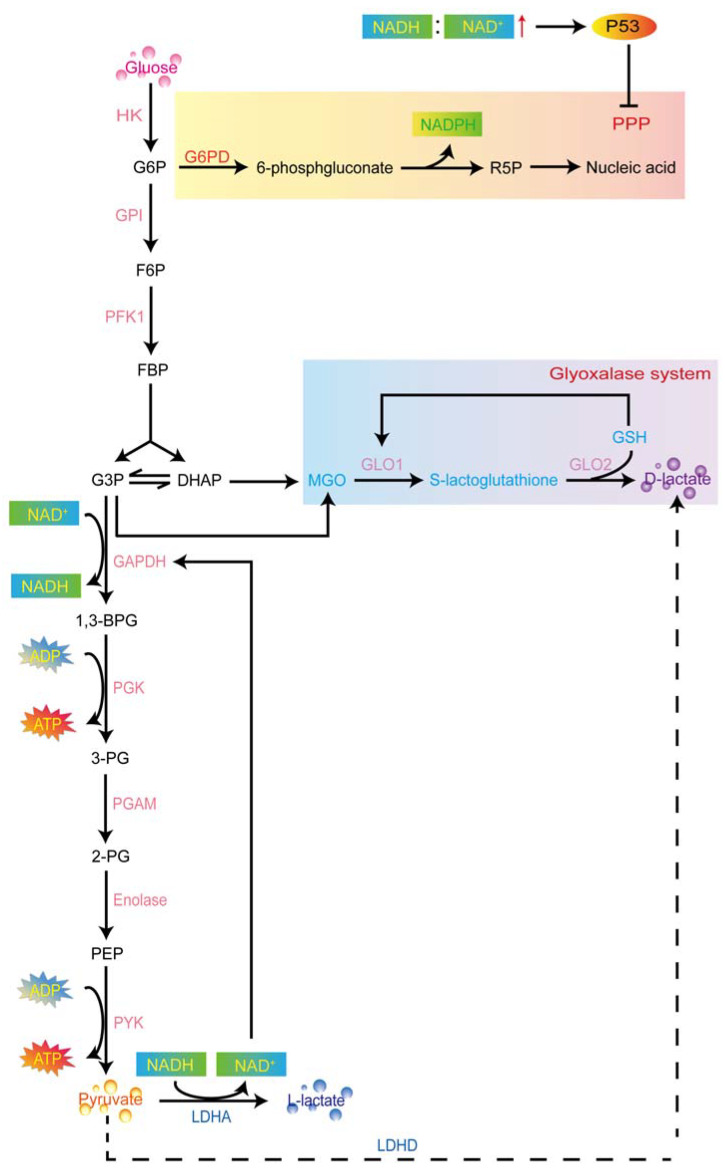
Lactate production in aerobic glycolysis. HK firstly catalyzes the glucose into GP6. G6PD and GPI convert G6P to 6-phosphgluconate and F6P, respectively. The 6-phosphgluconate finally produces the R5P for nucleotides synthesis which is known as the PPP. PFK1 catalyzes the F6P to FBP for pyruvate synthesis. L-lactate can be produced through the LDHA. FBP can also convert into DHAP and produce the intermediary product—MGO. Glyoxalases are involved in the detoxification of reactive MGO into D-lactate in a two-step reaction using GSH as a cofactor. HK, hexokinase; G6P, glucose 6-phosphate; G6PD, G6P dehydrogenase; GPI, G6P isomerase; F6P, fructose-6-phosphate; NADPH, nicotinamide adenine dinucleotide phosphate; R5P, ribose-5-phosphate; PPP, pentose phosphate pathway; NADH, reduced nicotinamide adenine dinucleotide; NAD^+^, nicotinamide adenine dinucleotide; PFK1, phosphofructokinase-1; FBP, fructose-1,6-bisphosphate; G3P, glyceraldehyde 3-phosphate; DHAP, dihydroxyacetone phosphate; GAPDH, G3P-dehydrogenase; 1,3-bisphosphoglycerate, 1,3-BPG; PGK, phosphoglycerate kinase; 3-PG, 3-phosphoglycerate; ATP, adenosine triphosphate; PGAM1, phosphoglycerate mutase 1; 2-PG, 2-phosphoglycerate; PEP, phosphoenolpyruvate; PYK, pyruvate kinase; LDHA, lactate dehydrogenase isoform A; MGO, methylglyoxal; GLO1, glyoxalase 1; GLO2, glyoxalase 2; GSH, glutathione.

**Figure 2 cancers-15-00087-f002:**
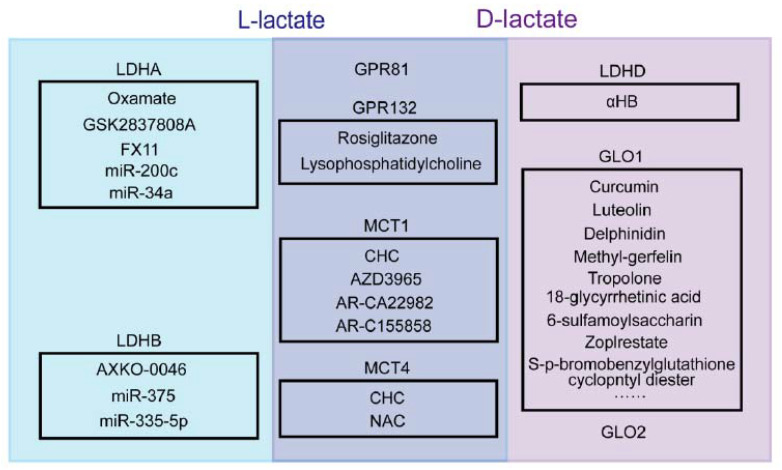
A graphic describes the lactate metabolism related molecular hallmarks and their inhibitory pharmacological agents or miRNAs in cancer. The L-lactate metabolism related enzymes are LDHA and LDHB (blue box). The D-lactate metabolism related enzymes are LDHD, GLO1, and GLO2 (purple box). GPR81, GPR132, MCT1, and MCT4 are affected by both the L- and D-lactate metabolism in cancer (bice box). LDHA, lactate dehydrogenase isoform A; LDHB, lactate dehydrogenase isoform B; LDHD, lactate dehydrogenase isoform D; GPR81, G-protein-coupled receptor 81; GPR132, G protein-coupled receptor 132; MCT1, monocarboxylate transporter 1; MCT4, monocarboxylate transporter 4; GLO1, glyoxalase 1; GLO2, glyoxalase 2; CHC, α-cyano-4-hydroxycinnamate; αHB, Alpha-hydroxy acid 2-hydroxy-3-butynoate; NAC, N-acetyl-cysteine.

**Figure 3 cancers-15-00087-f003:**
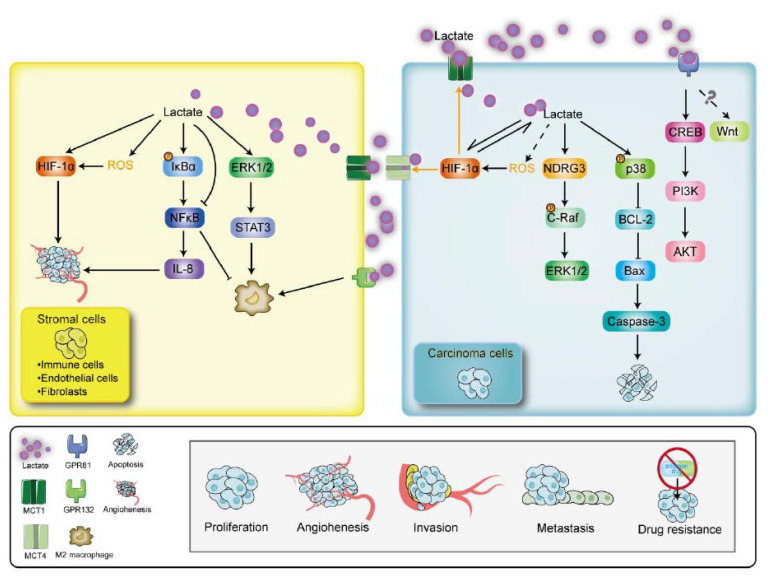
The downstream signalings of lactate mediation in cancer. Lactate production in the TME can mediate the signaling molecules in both stromal and carcinoma cells. In stromal cells, lactate can stimulate the HIF-1α, ROS, IκBα, and ERK1/2. In carcinoma cells, lactate can influence the HIF-1α, NDRG3, p38, and CREB signaling. HIF-1α, hypoxia induced factor-1 alpha; ROS, reactive oxygen species; NF-κB, nuclear factor kappa B; IL-8, interleukin-8; ERK1/2, extracellular signaling-regulated kinase 1/2; NDRG3, N-myc downstream-regulated gene family member 3; BCL-2, B-cell lymphoma-2; Bax, Bcl-2-associated X protein; CREB, cAMP response element binding protein; PI3K, phosphoinositide 3-kinase; AKT, protein kinase B.

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
