# Peer review of "Understanding the Contribution of Lactate Metabolism in Cancer Progress: A Perspective from Isomers"

_cancers, 2022, doi:10.3390/cancers15010087_

Round 1

Reviewer 1 Report

Well done.

-Cancer cells shift their metabolism to aerobic glycolysis to harvets the energy, and lactacte metabolism thereby plays a central role. The review deals with role and importance of understanding of lactate metabolism in cancer. There are several isomers of lactate (D- and L-lactate) that need to be taken into consideration when studying cancer. The cell physiological consequences of the shift are also discussed with regard to the review's focus.

-The authors illuminate the cell physiological and biochemical alterations and requirements for this rather complex shift of metabolism.

-This is a timely topic of particular interest for the broad cancer community.

-The review deals with several types of cancer (breast, cervical, lung, pancreatic, prostate and liver), and discusses the relevance of lactate metabolism for prognostic and therapeutic aspects related to cancer.
This information is well presented and it will benefit the cacer community.
-All figures are well presented and the references are appropriately selected and placed. 

Author Response

Thank you very much for your pertinent comments and valuable suggestions. According to the suggestion, we have revised and corrected the English language to increase the readability. Revised portion are marked in red in the paper.

Reviewer 2 Report

Manuscript ID: cancers-2064058

Type of manuscript: Review

Title: Understanding the contribution of lactate metabolism in cancer progress: a perspective from isomers

Authors: Ming Cai, Jian Wan, Keren Cai, Haihan Song, Yujiao Wang, Wanju Sun, Jingyun Hu

Comments

Although the authors drafted a concise review article on the “Understanding the contribution of lactate metabolism in cancer 2 progress: a perspective from isomers,” correct punctuation and grammar will add clarity and precision to writing and allows the reader to appreciate the content. The entire manuscript needs to be rewritten.

Line number 69-70: In fact, there are several potential advantages of glycolysis in carcinoma cells: glycolysis can synthesize ATP faster than aerobic oxidation to provide energy supply rapidly for the proliferation of carcinoma cells in spite of less efficient ATP production in this way (9); glycolysis reduces the reliance on oxygen for ATP production and thereby the potentially destructive reactive oxygen species (ROS) produced by the mitochondrial electron transport chain;

·                     glycolysis can synthesize ATP…… – does not make any sense

·                     the sentence ends with a semicolon instead of a period

Line number 97: Recently, lactate release by cancer cells will create a tumoral acidic microenvironment (21,22) and thereby promoting higher tumoral cell proliferation, survival, migration, invasion, and angiogenesis (6,22) and suppression of anticancer immune response (23).

·                     Recently, lactate release by cancer cells…… does not make sense

Reviewer 3 Report

In this paper, authors reviewed the metabolic pathway about lactate isomers production and revealed how lactate and its isomers benefit tumor growth. Which may provide guideline to target lactate metabolism for cancer therapy. Authors must address questions below before this paper can be accepted for publication.

1.     Line 38-40, please cite references here to prove high level of lactate in solid tumors is is associated with a poor prognosis.

2.     Line 46, author should cite reference#3 here again as this paper revealed that lactate can feed respiratory.

3.     Line 75, authors concluded here that Warburg effect can benefit biosynthesis, but they do not fully explain in above text how Warburg effect can do this. Except ATP synthesis, another major reason is that Warburg effect increased Pentose phosphate pathway, which provides precursors for lipid and nucleic acid synthesis. So please supplement this in above text.

4.     As author had mention pyruvate kinase in “2.2.1 L-lactate production in aerobic glycolysis”, please mark pyruvate kinase in Figure 1.

5.     In the section about lactate metabolism in specific cancer, authors just listed solid tumor types, breast cancer, cervical cancer, lung cancer, pancreatic cancer, prostate cancer and liver cancer. How does lactate metabolism affect leukemia?

Round 2

Reviewer 2 Report

The revised version is acceptable for publication.